

# Depression and insomnia among healthcare professionals during COVID-19 pandemic in Ethiopia: a systematic review and meta-analysis

Aragaw Asfaw Hasen[1], Abubeker Alebachew Seid[2] and Ahmed Adem Mohammed[2]

[1] Department of Statistics, College of Natural and Computational Sciences, Samara University, Semera, Afar, Ethiopia
[2] Department of Nursing, College of Medicine and Health Sciences, Samara University, Semera, Afar, Ethiopia

Corresponding author
Aragaw Asfaw Hasen,
aragawasfaw5@gmail.com

## ABSTRACT

**Introduction**. Healthcare professionals play a great role in the struggle against COVID-19. They are highly susceptible to COVID-19 due to their responsibilities. This susceptibility directly affects their mental health status. Comprehensive evidence on prevalence of depression and insomnia during this pandemic is vital. Thus, this study aims to provide the pooled prevalence of depression and insomnia, and their associated factors during the COVID-19 pandemic.

**Materials and methods**. This systematic review and meta-analysis follow the Preferred Reporting Items for Systematic Review and Meta-Analysis (PRISMA) guidelines. Studies were searched from PubMed, Cochrane Library, CrossRef, African Journals Online and Google Scholar databases from the occurence of the pandemic to June 2022. Study selection, data extraction and methodological quality assessment were done by two authors independently. The $I^2$ statistics was used for testing heterogeneity. A random effect model was used. Stata version 16.0 was used for statistical analysis.

**Results**. Eight studies were incorporated for this systematic review and meta-analysis. From seven studies the pooled prevalence of depression was 40% (95% CI [0.23–0.57]; $I^2 = 99.00\%$; $P = 0.00$). From three studies the pooled prevalence of insomnia was 35% (95% CI [0.13–0.58]; $I^2 = 98.20$; $P = 0.00$). Associated factors of depression on healthcare workers (HCWs) were being female pooled AOR: 2.09; 95% CI [1.41–2.76], been married (pooled AOR = 2.95; 95% CI [1.83–4.07]). Due to limited studies available on the factors associated with insomnia, it is impossible to pool and associated factors were presented in narrative synthesis.

**Conclusion**. COVID-19 is highly associated with the prevalence of depression and insomnia among healthcare professionals in Ethiopia. The pooled prevalence of depression and insomnia were significantly higher among healthcare professionals. Appropriate psychological counseling package should be realized for healthcare workers (HCWs) in order to recover the general mental health problems. Trial registration. This review was registered PROSPERO with registration number CRD42022314865.

## INTRODUCTION

The pandemic disease caused by severe acute respiratory syndrome coronavirus 2 (SARS CoV-2), first reported by officials in Wuhan City, China, in December 2019, and spread worldwide (*World Health Organization, 2020*). Healthcare professionals are forefront in the struggle against COVID-19. They are particularly susceptible to this disease due to their clinical task in the healthcare settings. This affects their mental health status. Studies have started investigating the mental health condition during COVID-19 pandemic. Study findings on Spanish healthcare workers (HCWs) imply that COVID-19 has impact on the mental health of HCWs (*García-Fernández et al., 2022*). In Iran, more than half of the nurses had depression in response to the COVID-19 outbreak (*Pourteimour, Yaghmaei & Babamohamadi, 2021*). In Africa, the prevalence of depression is higher compared to those reported elsewhere (*Chen et al., 2021*). The UNICEF situation report points Ethiopia had 96,169 confirmed cases of COVID-19 on 31 October, 2020 about 1,876 healthcare workers had tested positive and 77 had died (*UNICEF, 2020*).

Results of studies in Ethiopia on the prevalence of depression was reported as 66.4% (*Yadeta, Dessie & Balis, 2021*), 21.5% (*Wayessa et al., 2021*), 25.8% (*GebreEyesus et al., 2021*) and 58.2% (*Asnakew, Amha & Kassew, 2021*). Being female (*Yadeta, Dessie & Balis, 2021*), age and family size (*Wayessa et al., 2021*), with medical illness (*Wayessa et al., 2021*; *Asnakew, Amha & Kassew, 2021*), been married, being pharmacist, and contact with COVID-19 patients (*Asnakew, Amha & Kassew, 2021*) are factors associated with depressive symptom of healthcare professionals during the pandemic. Also studies showed the prevalence of insomnia was 15.9% (*Jemal et al., 2021*), 50.20% (*Yitayih et al., 2021*) and 40.8% (*Habtamu et al., 2021*). Being female, been married and working in emergency unit are factors associated with insomnia of healthcare workers during the pandemic (*Yitayih et al., 2021*). Furthermore, a worldwide meta-analysis during the pandemic result showed that the pooled prevalence of depression 34.31%. Mental health problems require early detection and initiation of intervention during the COVID-19 pandemic (*Necho et al., 2021*).

Study findings in Ethiopia on mental health problems (depression and insomnia) during the COVID-19 pandemic were varied (*Jemal, Deriba & Geleta, 2021*; *Wayessa et al., 2021*; *Jemal et al., 2021*; *Yadeta, Dessie & Balis, 2021*; *GebreEyesus et al., 2021*; *Asnakew, Amha & Kassew, 2021*; *Yitayih et al., 2021*; *Habtamu et al., 2021*). A comprehensive evidence on these findings helps policy makers, practitioners and researchers in numerous ways. This study aims to provide the pooled prevalence of depression and insomnia, and their associated factors among healthcare professionals during the COVID-19 pandemic in Ethiopia.

## MATERIALS AND METHODS

### Protocol registration

This study was conducted in accordance with the Preferred Reporting Items for Systematic Reviews and Meta-Analyses (PRISMA) statement and registered in the International

Prospective Register of Systematic Reviews with PROSPERO registration number: CRD42022314865.

## Search strategy

PubMed, Cochrane Library, CrossRef, African Journals Online and Google Scholar databases were searched for articles published from the occurence of the pandemic to June 2022. To assess the mental health impact of COVID-19 among healthcare professionals, observational studies were considered. Systematic searches were conducted by combining every possible combination of medical subject headings (MeSH) terms and keywords. Reference lists of key full text articles included in the review were checked to recognize any potentially eligible studies. The systematic procedure verifies that the literature search comprises all published studies on the impact of COVID-19 among healthcare professionals in Ethiopia. The search results were exported to Mendeley and duplicates were removed (*Kwon et al., 2015*). Two authors (AAH and AAS) independently screened titles and abstracts of the studies, and any disagreement between the authors was resolved by discussin with third author (AAM). The search strategy of PubMed database is presented (Table 1). The search strategy is considered as adequate to reduce the risk of selection and detection bias. For this study only observational studies (cohort, case-control and cross-sectional) focus the impacts of COVID-19 on depression and insomnia among healthcare professionals during the pandemic in Ethiopia were included.

**Setting/context:** Studies conducted in Ethiopia was the main concern of this review.

**Population:** All categories of healthcare professionals in Ethiopia.

**Study design:** Observational studies (cohort, case-control and cross-sectional studies) that reported the prevalence and associated factors of mental disorders among healthcare professionals during the COVID-19 pandemic.

**Language:** English language reported studies were considered.

The following types of studies were excluded: studies on whole populations; studies with very small sample size ($n < 30$); studies that did not have enough statistical information to be extracted and descriptive reviews, randomized controlled trials, systematic review, meta-analysis, editorials, comments, conference abstracts, and expert opinions, not exactly reported the prevalence and the determinants of mental health problems among healthcare professionals were excluded.

## Outcome measures

There are two main outcomes in this systematic review and meta-analysis. The first outcomes were the prevalence of depression and insomnia on healthcare professionals during the COVID-19 pandemic. The second outcome of the study was factors related to the prevalence of depression and insomnia among healthcare professionals during the COVID-19 pandemic in Ethiopia.

## Selection of studies

Two authors (AAH and AAS) assessed the studies based on inclusion and exclusion criteria. Firstly, the authors assessed both the titles and abstracts of the studies identified from the searched databases. Then full-text screening was done to screen the full texts selected in

**Table 1  PubMed search strategy.**

| Search number | Search detail |
|---|---|
| #1 | "COVID-19" [MeSH Terms] |
| #2 | "depression"[Mesh Terms] |
| #3 | "insomnia"[Mesh Terms] |
| #4 | "COVID-19" [Title/Abstract] OR "2019 novel coronavirus disease"[Title/Abstract] OR "2019 novel coronavirus infection"[Title/Abstract] OR "2019 ncov disease"[Title/Abstract] OR "2019 ncov infection"[Title/Abstract] OR "covid 19 pandemic"[Title/Abstract] OR "covid 19 pandemics"[Title/Abstract] OR "covid 19 virus disease"[Title/Abstract] OR "covid 19 virus infection"[Title/Abstract] OR "COVID19" [Title/Abstract] OR "coronavirus disease 2019" [Title/Abstract] OR "coronavirus disease 19" [Title/Abstract] OR "sars coronavirus 2 infection"[Title/Abstract] OR "sars cov 2 infection"[Title/Abstract] OR "severe acute respiratory syndrome coronavirus 2 infection""[Title/Abstract] OR "SARS-CoV-2" [Title/Abstract] OR "2019 novel coronavirus"[Title/Abstract] OR "2019 novel coronavirus"[Title/Abstract] OR "2019- nCoV"[Title/Abstract] OR "covid 19 virus"[Title/Abstract] OR "covid19 virus"[Title/Abstract] OR "Coronavirus disease 2019 virus"[Title/Abstract] OR "SARS coronavirus 2" [Title/Abstract] OR "SARS cov 2 virus"[Title/Abstract] OR "severe acute respiratory syndrome coronavirus 2" [Title/Abstract] OR "Wuhan coronavirus"[Title/Abstract] OR "Wuhan seafood market pneumonia virus"[Title/Abstract] |
| #5 | "Mental illness" [Title/Abstract] OR "Psychiatric problem" [Title/Abstract] AND "insomnia" [Title/Abstract] OR "depression" [Title/Abstract] OR "psychology problem" [Title/Abstract] OR "mental health effect" [Title/Abstract] OR "psychological disturbance" [Title/Abstract] "Mental Disorder" [Title/Abstract] OR "Psychiatric Illness" [Title/Abstract] OR "Psychiatric Diseases" [Title/Abstract] OR "Psychiatric Disorders" [Title/Abstract] OR Behavior Disorders" [Title/Abstract] OR "Severe Mental Disorder" [Title/Abstract] |
| #6 | "healthcare professionals"[Title/Abstract] OR "healthcare workers"[Title/Abstract] AND "Ethiopia"[Title/Abstract] OR "Addis Ababa"[Title/Abstract] OR "Amhara"[Title/Abstract] OR "Afar"[Title/Abstract] OR "Oromia"[Title/Abstract] OR "SNNP"[Title/Abstract] OR "Somali" [Title/Abstract] OR "Gambella" [Title/Abstract] OR " Benishangul-Gumuz" [Title/Abstract] OR "Tigrai" [Title/Abstract] OR " Harari" [Title/Abstract] OR "Dire Dawa" [Title/Abstract] |
| #7 | #1 OR #4 |
| #8 | #2 OR #3 OR #5 |
| #9 | #6 AND #7 AND #8 |
| #10 | Limit to "observational studies" OR "cohort" OR "case-control" OR "cross-sectional" |

the previous stage. Moreover, we have a rationale for inclusion and exclusion of studies in the PRISMA flow diagram. Lastly, the final list of articles for data extraction for systematic review and meta-analysis was prepared.

## Data extraction

The following data were extracted from each article by two authors independently: author's name, study type, total number of participants, year of publication, region, study design cases, sample size, instrument used, mental disorders, prevalence of mental disorders, and significant associated factors of mental disorders with their effect size. There was pretest the data extraction form to ensure effective, facilitates the collection of all necessary data required for the valuable systematic review and meta-analysis. Disagreements were resolved by deep argument among authors.

## Methodological quality assessment

Two authors (AAH and AAS) separately assessed the quality of included studies using the Newcastle-Ottawa Scale (NOS) (*Stang, Jonas & Poole, 2018*). The scale scores observational studies based on three parameters: selection, comparability and exposure/outcome
assessment. Studies with less than five scores were considered low quality, five to seven scores of moderate quality, and more than seven scores of high quality (*Ssentongo, Ssentongo & Heilbrunn, 2020*). Only studies with moderate and above quality score were included in this systematic review and meta-analysis.

## Data synthesis

The extracted data was entered into a Microsoft Excel and then imported in to Stata version 16.0 (StataCorp. 2019. Stata Statistical Software: Release 16. College Station, TX: StataCorp LLC) software for the analyses. We calculated pooled prevalence and pooled adjusted odds ratios (AOR) with 95% confidence interval (CI) by the generic inverse variance method. Heterogeneity among included studies was assessed using the $I^2$ test. If $I^2 > 0.5$ or $P < 0.1$ it is considered that there is a significant heterogeneity among the included studies (*Zhu et al., 2020*) and random-effect model with the inverse variance method was used. To determine the source of heterogeneity subgroup analyses was performed by regions and instruments used in individual studies, the difference between the subgroup was assessed by Cochran's $Q$-statistics (*Bowden et al., 2011*).

# RESULTS

A PRISMA flow diagram illustrating the steps of data search and refining process for the study on depression and insomnia among healthcare professionals during the COVID-19 pandemic period (Fig. 1). We have got 30 papers from the searched databases. Eight studies duplicated were removed, we examined the titles and abstracts and three papers were removed. By examining the full text, we removed four that did not meet inclusion criteria. Seven Studies were removed due to not reporting about depression and insomnia. Finally, eight studies were relevant to the systematic review and meta-analysis.

## Characteristics of included studies

In this systematic review and meta-analysis, we included 8 studies (*Jemal, Deriba & Geleta, 2021*; *Wayessa et al., 2021*; *Jemal et al., 2021*; *Yadeta, Dessie & Balis, 2021*; *GebreEyesus et al., 2021*; *Asnakew, Amha & Kassew, 2021*; *Yitayih et al., 2021*; *Habtamu et al., 2021*) focusing on the impact of COVID-19 on depression and insomnia among healthcare professionals in Ethiopia. Regarding the regional distribution one study (*GebreEyesus et al., 2021*) is from SNNP, three studies (*Jemal, Deriba & Geleta, 2021*; *Wayessa et al., 2021*; *Yadeta, Dessie & Balis, 2021*) are from Oromiya, one study (*Asnakew, Amha & Kassew, 2021*) is from Amhara, one study (*Jemal et al., 2021*) is from Addis Ababa and Oromiya, one study (*Habtamu et al., 2021*) is from Addis Ababa. Furthermore, the key characteristics of the included papers was summarized and showed in table (Table 2).

## Quality of included studies

The methodological quality score of the eight included studies using the modified Newcastle Ottawa scale for correctional studies quality assessment tool was presented (Table 2). Accordingly, two studies were rated as moderate quality (*Asnakew, Amha & Kassew, 2021*; *Yitayih et al., 2021*) and six studies were rated as high quality (*Jemal, Deriba & Geleta, 2021*;

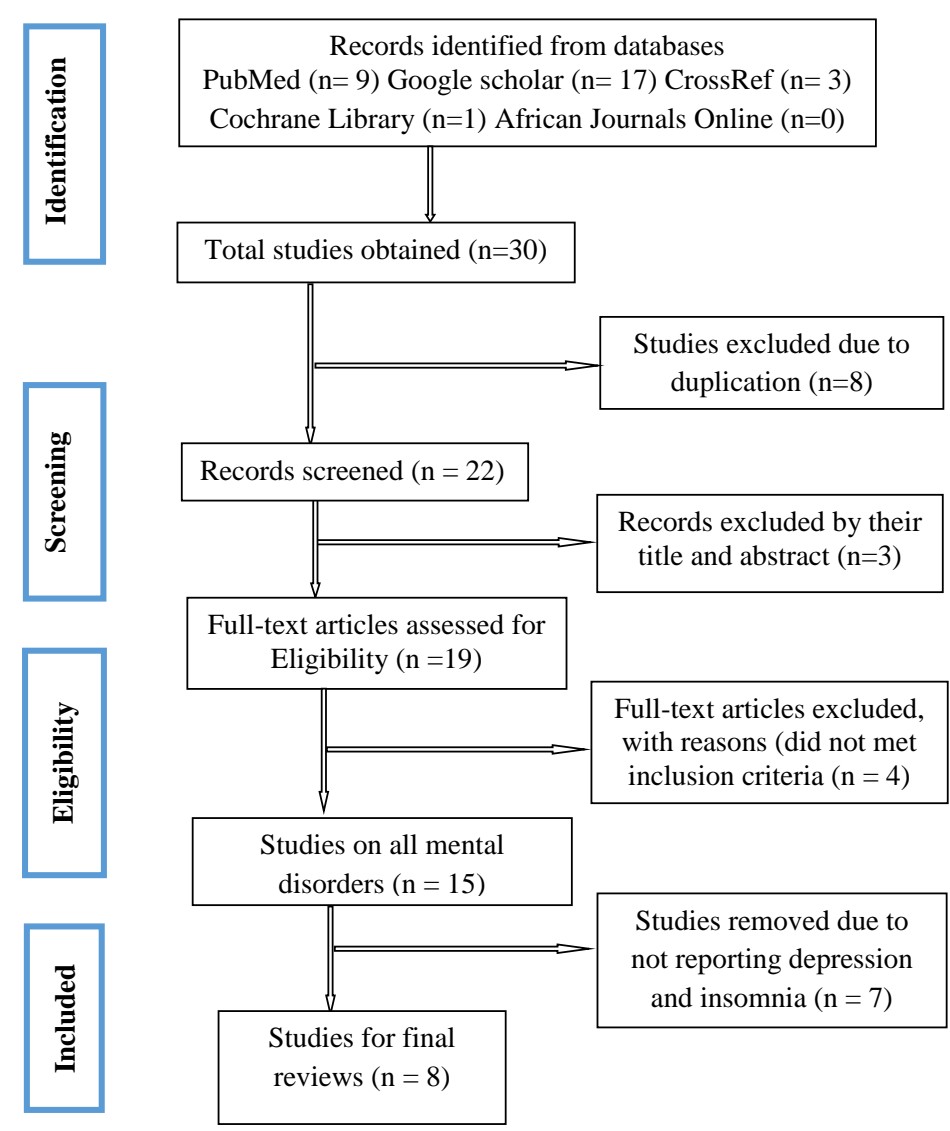

**Figure 1** Preferred reporting items for systematic reviews and meta-analyses (PRISMA) flowchart.

*Jemal et al., 2021*; *Yadeta, Dessie & Balis, 2021*; *GebreEyesus et al., 2021*; *Habtamu et al., 2021*; *Wayessa et al., 2021*) were considered for final systematic review and meta analysis.

## Publication bias

Detection of publication and related biases is vital for the validity and interpretation of meta-analytical findings. The test power is usually too low to distinguish chance from real asymmetry when there are less than 10 studies in the meta-analysis (*Furuya-Kanamori, Barendregt & Doi, 2018*). Accordingly, the number of included studies for depression and insomnia are less than ten we do not apply the asymmetry test.

**Table 2   Key characteristics of the included studies for depression and insomnia of HCWs during the COVID-19 pandemic in Ethiopia.**

| No | Authors (year) | Region | Study design | Gender (male %) | Cases | n | Mental disorders | Instrument | P (%) | Quality |
|----|----------------|--------|--------------|-----------------|-------|---|------------------|------------|-------|---------|
| 1 | *Jemal et al. (2021a)* | AA and Oromiya | CS | 540 (66.17) | 492 | 816 | Depression | DASS-21 | 60.3 | 8 |
| 2 | *Jemal, Deriba & Geleta (2021b)* | Oromiya | CS | 279 (66.90) | 66 | 417 | Insomnia | ISI | 15.9 | 8 |
| | | | | | 68 | 417 | Depression | PHQ-9 | 16.3 | |
| 3 | *Yitayih et al. (2021)* | Oromiya | CS | 118 (47.38) | 125 | 249 | Insomnia | ISI | 50.2 | 7 |
| 4 | *GebreEyesus et al. (2021)* | SNNP | CS | 167 (51.86) | 83 | 322 | Depression | PHQ-9 | 25.8 | 9 |
| 5 | *Habtamu et al. (2021)* | AA | CS | 101 (42.43) | 65 | 238 | Depression | PHQ-9 | 27.3 | 9 |
| | | | | | 97 | 238 | Insomnia | PSQI | 40.8 | |
| 6 | *Wayessa et al. (2021)* | Oromiya | CS | 173 (62.90) | 59 | 275 | Depression | DASS-21 | 21.5 | 8 |
| 7 | *Yadeta, Dessie & Balis (2021)* | Oromiya | CS | 133 (50.18) | 176 | 265 | Depression | PHQ-9 | 66.4 | 8 |
| 8 | *Asnakew, Amha & Kassew (2021)* | Amhara | CS | 292 (69.7) | 244 | 419 | Depression | DASS-21 | 58.2 | 7 |

**Notes.**

P, prevalence; *n*, sample size; CS, cross-sectional; DASS-21, 21-item Depression Anxiety Stress Scale; PHQ-9, 9-item Patient Health Questionnaire; ISI, Insomnia Severity Index; PSQI, Pittsburgh Sleep Quality Index; AA, Addis Ababa; SNNP, Southern nations nationalities and people.

## Pooled prevalence of depression

A total of seven studies reported the prevalence of depression, and the pooled prevalence of the depression was 40% (95% CI [0.23–0.57]; $I^2 = 99.00\%$; $p = 0.00$) (Fig. 2). From the heterogeneity test, there is significant heterogeneity is observed among individual studies on the prevalence of depression among healthcare professionals during the pandemic in Ethiopia.

## Subgroup analysis of depression by region

To handle the variability in studies the subgroup analysis by region is done. From the forest plot (Fig. 3), the pooled prevalence of depression in Addis Ababa & Oromiya, Oromiya, SNNP, Addis Ababa and Amhara is 60%, 35%, 26%, 27% and 63% respectively. The heterogeneity test ($Q = 226.85$, $p = 0.000$) indicates that there is significant variability among regions. The prevalence of depression is higher in Amhara region compared to the others.

## Subgroup analysis of depression by instrument

Based on the instrument used in individual included studies, subgroup analysis is done. From the forest plot (Fig. 4), the pooled prevalence of depression by DASS-21 and PHQ-9 is 48%, and 34% respectively. The heterogeneity test ($Q = 0.79$, $p = 0.374$) indicates that there is no significant variability on a study finding between measurements. The prevalence of depression measured in DASS-21 is higher than that measured by PHQ-9 among healthcare workers during the pandemic. This might be due to the difference in sensitivity and specificity of the assessment instruments.

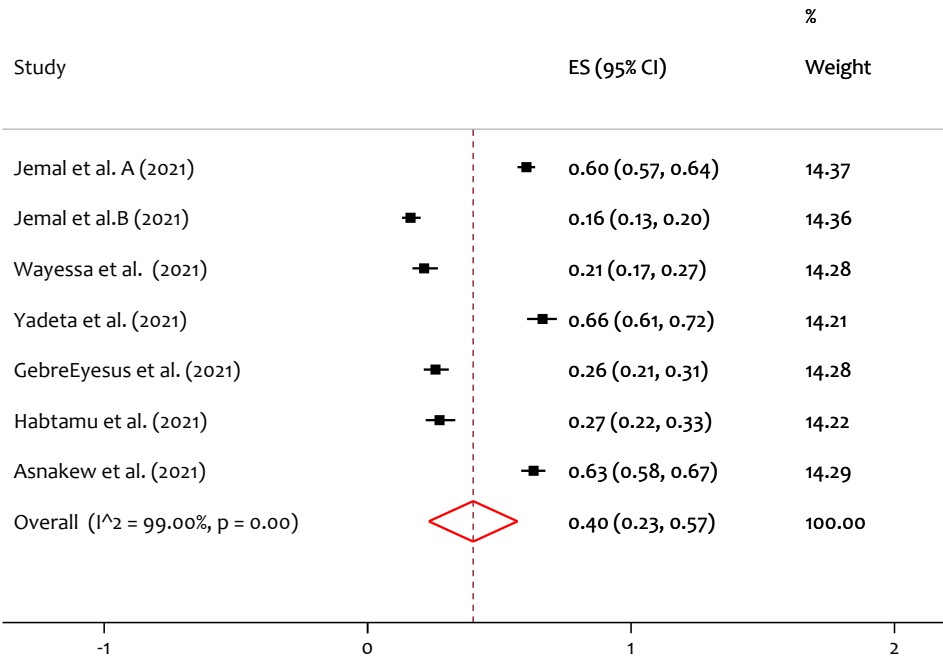

| Study | | ES (95% CI) | % Weight |
|-------|-----|-------------|--------|
| Jemal et al. A (2021) | | 0.60 (0.57, 0.64) | 14.37 |
| Jemal et al.B (2021) | | 0.16 (0.13, 0.20) | 14.36 |
| Wayessa et al. (2021) | | 0.21 (0.17, 0.27) | 14.28 |
| Yadeta et al. (2021) | | 0.66 (0.61, 0.72) | 14.21 |
| GebreEyesus et al. (2021) | | 0.26 (0.21, 0.31) | 14.28 |
| Habtamu et al. (2021) | | 0.27 (0.22, 0.33) | 14.22 |
| Asnakew et al. (2021) | | 0.63 (0.58, 0.67) | 14.29 |
| Overall (I^2 = 99.00%, p = 0.00) | | 0.40 (0.23, 0.57) | 100.00 |

**Figure 2** **Forest plot for the prevalence of depression among the healthcare professionals during COVID-19 pandemic.** ES, effect size; CI, confidence interval; Weight, weight of each included study.

## Pooled prevalence of insomnia

Three studies reported the prevalence of insomnia, and the pooled prevalence of the insomnia was 35% (95% CI [0.13–0.58]; $I^2 = 98.20\%$; $p = 0.00$) (Fig. 5). In the test of heterogeneity, we have seen that there is considerable variation among individual included studies on the prevalence of insomnia among healthcare professionals during the pandemic in Ethiopia.

## Subgroup analysis of insomnia by region

Subgroup analysis by region is done. From the forest plot (Fig. 6), the pooled prevalence of insomnia in Oromiya and Addis Ababa is 24% and 41% respectively. The heterogeneity test ($Q = 22.01$, $p = 0.000$) indicates that there is significant variation in the prevalence of insomnia between regions. The prevalence of insomnia is higher in Addis Ababa than Oromiya.

## Subgroup analysis of insomnia by instrument

Based on the instrument used in individual included studies, subgroup analysis is done. From the forest plot in (Fig. 7), the pooled prevalence of depression measured by ISI and PSQI is 24% and 41% respectively. The heterogeneity test ($Q = 22.02$, $p = 0.000$) indicates that there is significant variation on a study finding between measurements. The prevalence of depression measured in PSQI is higher than that measured by ISI among HCWs during the pandemic.

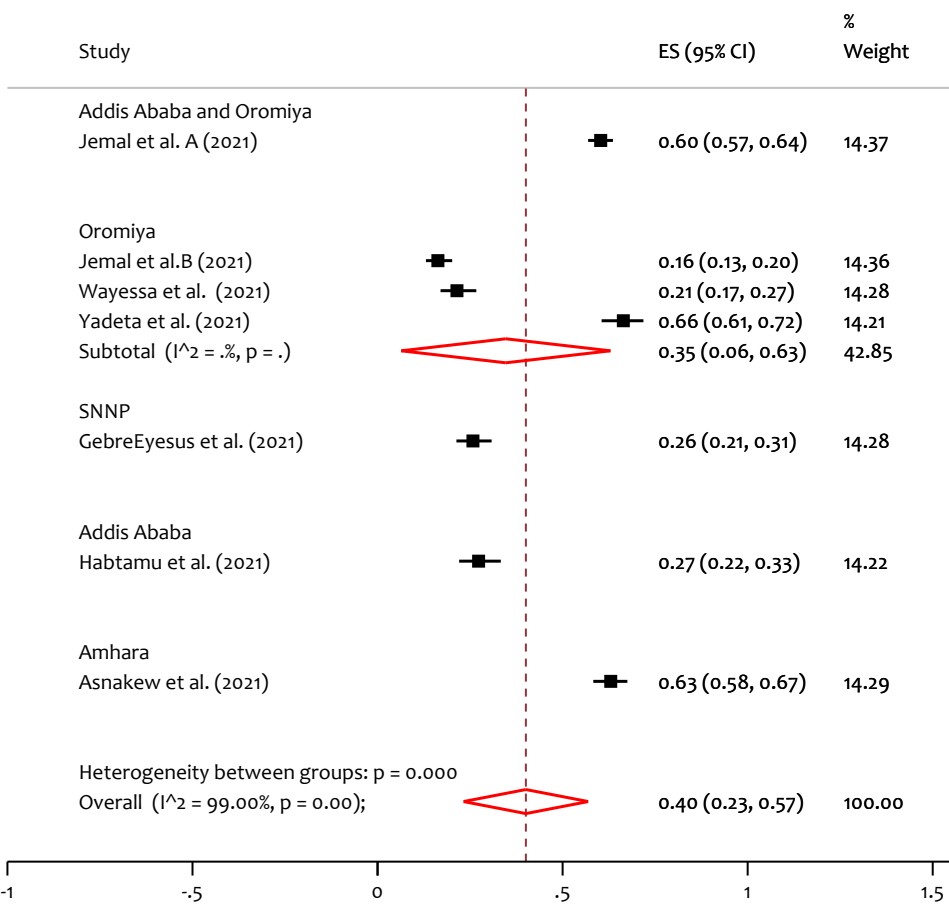

**Figure 3** **Subgroup analysis of prevalence of depression among HCWs during the COVID-19 pandemic by region.** SNNP, Southern nation's nationalities and people; ES, effect size; CI, confidence interval; Weight, weight of each included study.

## Pooled adjusted odds ratio of associated factors of depression and insomnia

The pooled adjusted odds ratio of the factors associated with prevalence of depression among healthcare professionals during COVID-19 in Ethiopia presented (Table 3).

The pooled adjusted odds ratio on female healthcare workers is 2.09, 95% CI [1.41–2.76], implies that the odds of female healthcare workers is two times more to develop the depressive symptom than males during the pandemic. Similarly for marital status (being married) the pooled adjusted odds ratio is 2.95, 95% CI [1.83–4.07] indicates that the odds of married healthcare workers is nearly 3 times more to develop depression than not married. Whereas, in this study working unit (isolation center) and with medical illness are not statistically significant variables on affecting the prevalence of depression during the pandemic. Since the findings on the factors associated with the prevalence of depression and insomnia were heterogeneous and limited, it is impossible to pooling. We explore and present these factors systematically as summarized (Table 4).
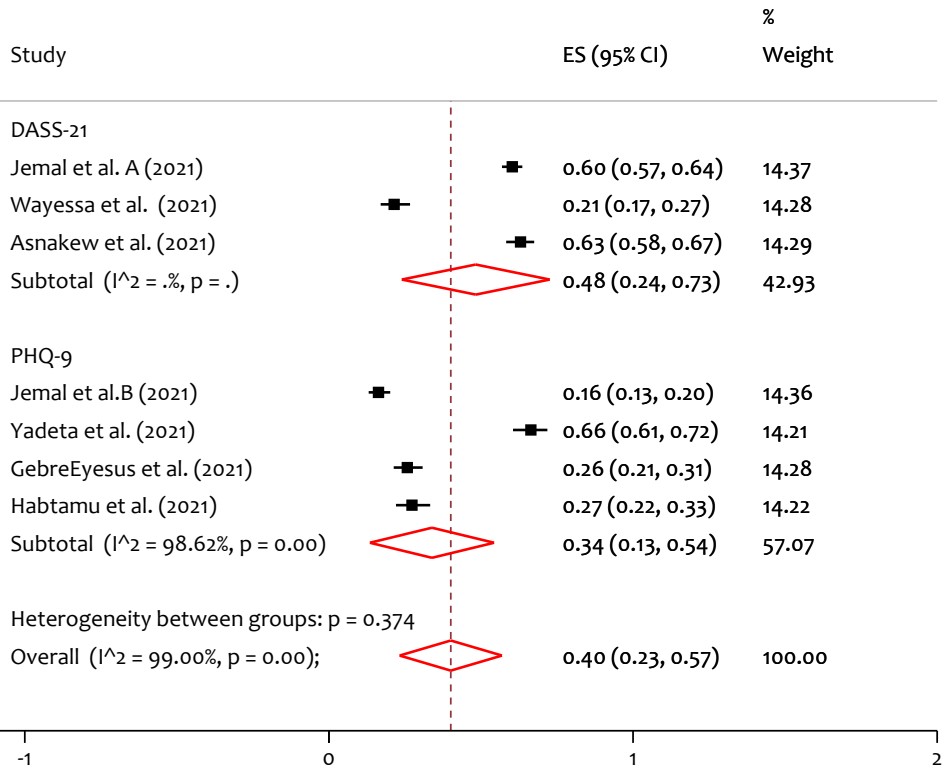

**Figure 4** Subgroup analysis for prevalence of depression among healthcare professionals during the COVID-19 pandemic in Ethiopia by instrument. DASS-21, Depression, Anxiety, Stress Scale-21; PHQ-9, the 9-item Patient Health Questionnaire.

## DISCUSSION

This study aims to investigate the pooled prevalence and associated factors of depression and insomnia among healthcare professionals during the COVID-19 pandemic in Ethiopia. The result shows a high prevalence depression and insomnia among healthcare professionals during the COVID-19 pandemic in Ethiopia. Consequently, there is a major concern for the mental health of HCWs during the COVID-19 pandemic, as well as in potential future public health crises. There are studies at the single level, but to our knowledge, this systematic review and meta-analysis study is the first of its kind that assessed the pooled prevalence of depression and insomnia and their associated factors.

The study included eight studies (*Jemal, Deriba & Geleta, 2021*; *Wayessa et al., 2021*; *Jemal et al., 2021*; *Yadeta, Dessie & Balis, 2021*; *GebreEyesus et al., 2021*; *Asnakew, Amha & Kassew, 2021*; *Yitayih et al., 2021*; *Habtamu et al., 2021*) focused on the impact of COVID-19 on depression and insomnia among healthcare professionals in Ethiopia. This reflects that the impact of COVID-19 on mental health problems especially depression and insomnia were not well investigated. With the available evidence the pooled prevalence of depression and insomnia, and their associated factors were discussed.
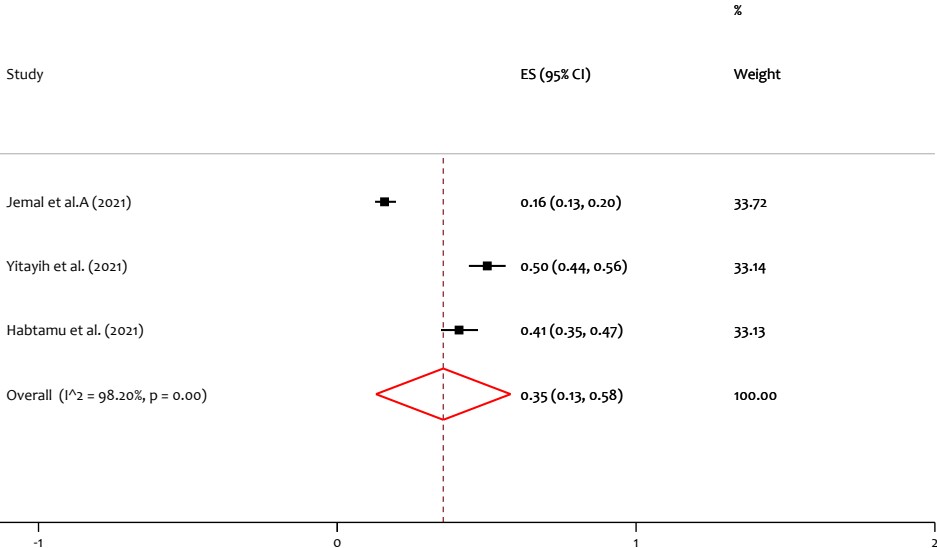

**Figure 5 Forest plot for the prevalence of insomnia among the healthcare professionals during the COVID-19 pandemic.** ES, effect size; CI, confidence interval; Weight, weight of each included study.

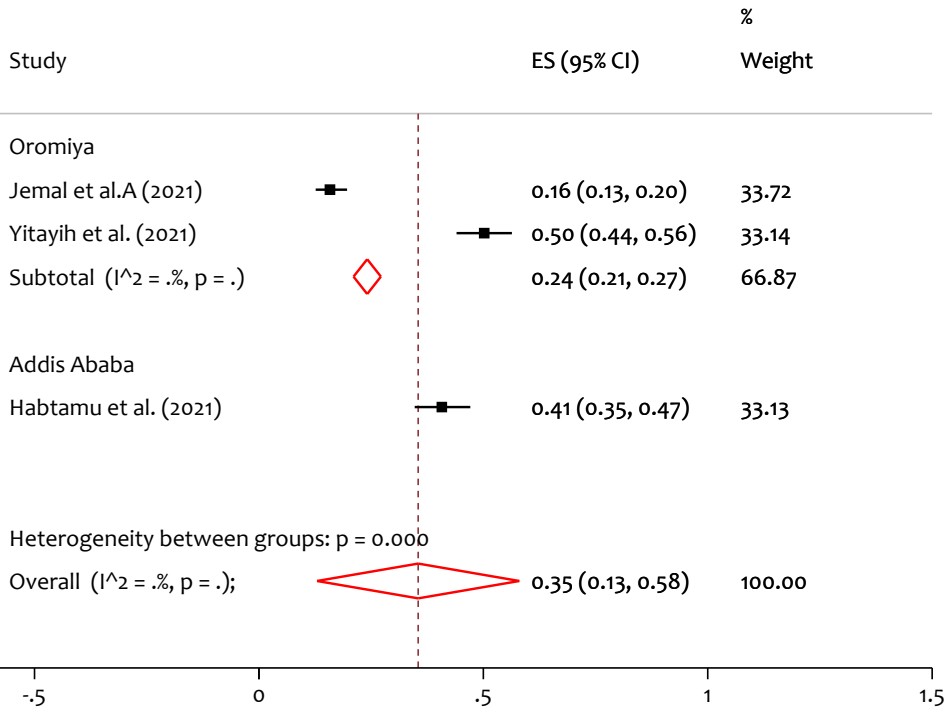

**Figure 6 Subgroup analysis of prevalence of insomnia among healthcare professionals during COVID-19 pandemic by region.** ES, effect size; CI, confidence interval; Weight, weight of each included study.

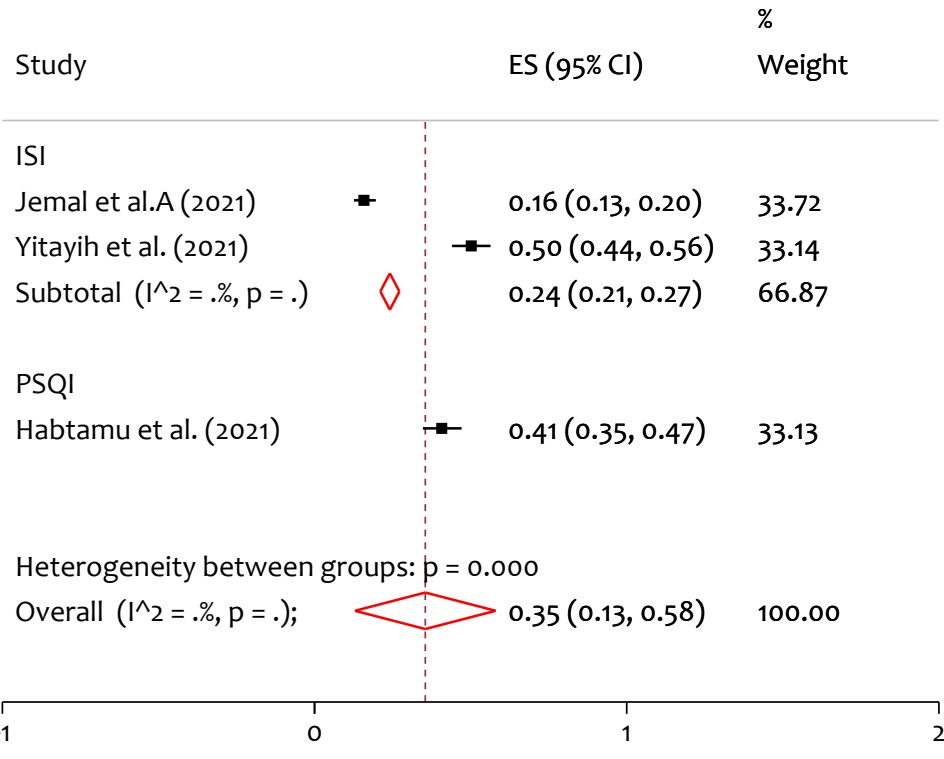

**Figure 7** **Subgroup analysis of prevalence of insomnia among healthcare professionals during the COVID-19 pandemic in Ethiopia by instrument.** ISI, Insomnia Severity Index; PSQI, Pittsburgh Sleep Quality Index. ES, effect size; CI, confidence interval; Weight, weight of each included study.

In this study the pooled prevalence of depression among HCWs during the COVID-19 pandemic was 40%. This is higher than the previous meta-analysis result of 36% (*Sun et al., 2021*), 37.12% (*Mahmud et al., 2021*), 31.8% (*Batra et al., 2020*), 26.2% (*Zhang et al., 2021*) and 31.1% (*Marvaldi et al., 2021*). However, the results is lower than the study results in Africa which is 45% (*Chen et al., 2021*) and in Kenya which is 45.9% (*Ali, Shah & Talib, 2021*).

Among regions, there is heterogeneity on the prevalence of depression. We found the prevalence of depression in Amhara regions is 63%, is higher compared to the others. This might be due to the difference in the availability of prevention equipment's for the COVID-19 and the levels of awareness on the pandemic. Also subgroup analysis by instrument, the pooled prevalence of depression by DASS-21 is 48%, this is higher than study in global 34.83% (*Mahmud et al., 2021*) and pooled result by PHQ-9 is 34%, lower than study meta analysis result 38.11% (*Mahmud et al., 2021*).

The pooled adjusted odds ratio on female healthcare workers is 2.09, 95% CI [1.41–2.76], implies that odds of female healthcare workers is two times more to develop the depressive symptom than males during the pandemic. This is inline with the study finding in Kenya females HCWs experiencing more symptoms of all the mental health disorders than males (*Ali, Shah & Talib, 2021*) and Egypt (*Elgohary et al., 2021*). Similarly for marital status

**Table 3  Pooled adjusted odds ratio of associated factors of depression.**

| Mental illness | Numbers of studies | Variables | Reference category | Pooled AOR (95% CI) | Heterogeneity | |
|---|---|---|---|---|---|---|
| | | | | | $I^2$ (%) | p-value |
| Depression | 4 | Sex (female) | Male | 2.09 (1.41, 2.76) | 0.00 | 0.837 |
| Depression | 2 | Working unit (COVID-19 isolation center) | Pharmacy | 2.13 (0.94, 3.31) | 0.00 | 0.980 |
| Depression | 3 | Marital status (married) | Single | 2.95 (1.83, 4.07) | 0.00 | 0.743 |
| Depression | 2 | With medical illness | Not | 4.11 (−1.66, 9.87) | 40.4 | 0.195 |

**Table 4  A summarized review of study findings on factors of depression and insomnia with their magnitude among healthcare professionals during the COVID-19 pandemic in Ethiopia.**

| Authors (Year) | Mental disorder | Variables | Category | AOR (95% CI) |
|---|---|---|---|---|
| *Jemal et al. (2021a)* | Depression | Hcws in the Oromiya zone | Centeral Oromiya | 3.94 (1.94, 8.09) |
| | | Medical laboratory professionals | Pharmacy | 4.69 (2.81, 9.17) |
| *Jemal, Deriba & Geleta (2021b)* | Depression | Married participants | Single | 2.87 (2.03, 4.30) |
| | | Emergency unit | Outpatient | 2.11 (1.27, 4.61) |
| | | Experience of <5 years | >=10 years | 2.07 (1.89, 4.84) |
| | | Poor behavioral responses | Good response | 2.13 (1.18, 3.57) |
| | | Poor perception to COVID-19 | Good | 1.47 (1.88, 2.64) |
| *GebreEyesus et al. (2021)* | Depression | Masters and above | Deploma | 10.844 (1.131,4.551) |
| | | Whose educational status, degrees | Deploma | 2.269 (3.314,35.482) |
| | | Live with their husband/wife | Alone | 5.824 (1.896,17.88) |
| | | Live with their families | Alone | 3.938 (1.380,11.242) |
| *Wayessa et al. (2021)* | Depression | Age 25–29 | Age >=35 | 2.35 (1.126,3.95) |
| | | Family size >=4 members | 1 person | 3.56 (1.09,11.62) |
| | | Alcohol use | Not | 4.31 (1.76, 10.55) |
| | | Having training on COVID-19 | Not | 0.37 (0.17–0.81) |
| | | Poor knowledge on COVID-19 | Good | 15.34 (6.32–37.21) |
| *Yadeta, Dessie & Balis (2021)* | Depression | Perceived susceptibility to COVID-19 | Not | 4.05 (1.12–14.53) |
| *Asnakew, Amha & Kassew (2021)* | Depression | With Mental illness | Not | 2.72 (1.05,7.01) |
| | | Contact confirmed COVID-19 patients | Not contct | 2.59 (1.37,4.89) |
| | | Poor social support | Good | 1.87 (1.08,3.22) |
| *Jemal, Deriba & Geleta (2021b)* | Insomnia | Female HCWs | Male | 2.16 (1.58, 4.38) |
| | | Married participants | Single | 3.31 (1.56, 6.68) |
| | | Working in the emergency units | Outpatient | 2.74 (1.85, 6.45) |
| | | Working experience of <5 years | >=10 years | 2.45 (1.28, 4.90) |
| | | Poor behavioral responses to COVID-19 | Good | 1.69 (1.02, 3.17) |
| | | Have poor perception COVID-19 | Good | 1.98 (1.56, 3.95) |

Notes.
AOR, Adjusted Odds Ratio; CI, Confidence Interval; HCWs, healthcare Workers.

(being married) the pooled adjusted odds ratio is 2.95, 95% CI [1.83–4.07] indicate odds of married healthcare workers are nearly 3 times more to develop depression than non married. Whereas, in this study working unit (isolation center) and with medical illness

are not statistically significant variables on affecting the prevalence of depression during COVID-19 pandemic.

The pooled prevalence of insomnia among HCWs during the COVID-19 pandemic was 35%. This is in line with the pooled prevalence of insomnia among healthcare workers in China which is 34.5% (*Zhang et al., 2021*) and in Kenya which is 37.0% (*Ali, Shah & Talib, 2021*). On the other hand the prevalence is higher compared to meta analysis results 28% (*Chen et al., 2021*), and 27.8% (*Batra et al., 2020*) however lower than the global meta analysis result 43.76% (*Mahmud et al., 2021*). Up on subgroup analysis by region, the heterogeneity test indicates that there is significant variation in the prevalence of insomnia between regions. The prevalence of insomnia in Addis Ababa is 41% higher than Oromiya. The pooled prevalence of depression measured by ISI is 24%, this is lower than study in China 36.1% (*Zhang et al., 2020*) similarly using PSQI is 41%, this is higher than study in Kenya 24.2% (*Kwobah et al., 2021*). The heterogeneity test indicates that there is significant variation on a study finding between measurements. This might be due to the sensitivity and specificity on the measurement tools. For insomnia, pooling the adjusted odds ratio for associated factors was not performed due to the limited data available.

This study is with strengths and some limitations. Study selection, data extraction and quality assessment were performed by two authors independently. Newcastle-Ottawa Scale used to assess the quality of the included studies were the strengths. Whereas, the absence of sufficient studies investigating the impact of COVID-19 on depression and insomnia among healthcare professionals in Ethiopia and heterogeneity among studies were the limitations of this systematic review and meta-analysis.

## CONCLUSION

The COVID-19 pandemic caused a variety of mental health impacts among healthcare professionals in Ethiopia. Due to this pandemic, the prevalence of depression and insomnia among healthcare professionals became high in Ethiopia. The prevalence varied among regions according to the instruments used. The suitable programs that offer awareness on the COVID-19 pandemic, psychological counseling and intervention should be implemented for HCWs to improve the general mental health problems including depression and insomnia.

### Abbreviations

| | |
|---|---|
| **AOR** | Adjusted odds ratio |
| **CI** | Confidence interval |
| **HCWs** | healthcare workers |
| **MeSH** | Medical subject headings |
| **NOS** | Newcastle Ottawa quality assessment scale |
| **PRISMA** | Preferred Reporting Items for Systematic Review and Meta-Analysis |
| **SNNP** | Southern Nations, Nationalities and People |
| **WHO** | World Health Organization |

## ACKNOWLEDGEMENTS

We acknowledged the authors of studies included in this systematic review and meta-analysis.

### Funding

The author received no funding for this work. The funders had no role in study design, data collection and analysis, decision to publish, or preparation of the manuscript.

### Competing Interests

The authors declare there are no competing interests.

### Author Contributions

- Aragaw Asfaw Hasen conceived and designed the experiments, performed the experiments, analyzed the data, prepared figures and/or tables, authored or reviewed drafts of the article, and approved the final draft.
- Abubeker Alebachew Seid conceived and designed the experiments, performed the experiments, analyzed the data, prepared figures and/or tables, authored or reviewed drafts of the article, and approved the final draft.
- Ahmed Adem Mohammed conceived and designed the experiments, performed the experiments, analyzed the data, prepared figures and/or tables, authored or reviewed drafts of the article, and approved the final draft.

### Data Availability

This is a systematic review and meta-analysis.

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
