# Peer review of "Depression and insomnia among healthcare professionals during COVID-19 pandemic in Ethiopia: a systematic review and meta-analysis"

_PeerJ, doi:10.7717/peerj.15039_

## Round 0.1 · original submission · Major Revisions

Please address to all the comments raised by the reviewers.

Reviewer 1 ·

Basic reporting

The article is well written, but it needs improving in some parts.
The authors must complete the statement in Introduction part with appropriate reference(s). (Line 71-74)
Some parts in Introduction should be deleted. (Line 77-91)
There are also some typos. Please check the whole sentences to prevent the typos.
Some references need changing because of more than 10 years (Line 323-380)

Experimental design

Line Section Comment
35 Abstract The authors should mention detail of Stata 16.0

39 Abstract Please write the full word of HCWs then write only the abbreviation
58 Introduction Please write the full word of HCWs then write only the abbreviation
71-74 Introduction Please the authors should give one or two references to support this statement
77-91 Research questions I think the authors should remove this part
93 Protocol Registration The authors must crosscheck with the arrangement of PRISMA research method
108-109 Search strategy Please add reference (s) to support that Mendeley can remove the duplicate data
118 Population What does this statement mean? Does it mean that the authors involve the population from healthcare professionals only in the Ethiopia or not? It seems ambiguous
119-120 Study design Does it mean that impact of mental disorders from the health workers or anything else?
123-130 Exclusion criteria It is better with the author write in the sentences not using the whole number
125 Exclusion criteria What does it mean? What is the range of very small sample size? It is not clearly mentioned
132-136 Outcome measures What is the urgency to analyze and identify this prevalence of depression and insomnia?
138 Selection of studies It is enough only to use two reviewers? Better the reviewers’ number are odd not in even numbers.
139-143 Selection of studies How can the authors ensure that it is enough only using two reviewers?
145 Data extraction How can the authors ensure this extraction process?
How can the authors ensure the eligibility from this data extraction process?
180 Study characteristics It is better using the participants characteristics or characteristic of included studies
251 Discussions The authors should mention that why the authors analyze this phenomenon? What is the most urgency so that authors analyze this prevalence of depression and insomnia?
255 Discussions Is it enough to use only 8 studies? These 8 studies are counted from the minimal journal that have to be analyzed?
277 Discussions How the authors can ensure the eligibility of these performances?
290 Abbreviations The authors should write the stands for all of abbreviations first then only write the abbreviations.
323 References The authors shoud match writing the reference style with the template or published manuscript in PeerJ
331 References I think this reference is incomplete because of lacking information about volume, issue, and page. It is better to the authors to repair this.
333 References Does the author mean UNICEF? I searched the word UNICF but there was no UNICF. Please repair this if authors made typo.
362 References This reference is more than 5 years. It needs changing
372 References This reference needs changing because of more than 10 years
380 References This reference needs changing because of more than 10 years
- Table 2 Please use the function “distribute coloumn” so that the title of table can be directly seen because the former table title is difficult to understand.
Example: study design not stud-y desi-gn
The parts that need changing have been marked with comment in the whole manuscript.
- Figure 1 until 6 Please improve the quality of the figure. It seems so blur.

Validity of the findings

-

Additional comments

-

Annotated reviews are not available for download in order to protect the identity of reviewers who chose to remain anonymous.

·

Basic reporting

Clear and unambiguous

Experimental design

Sentence: To understanding of impacts of COVID-19 on depression and insomnia among health care professionals in Ethiopia, we conducted a systematic review and meta-analysis in accordance with the Preferred Reporting Items for Systematic Reviews and Meta-Analyses (PRISMA) statement and registered in the International Prospective Register of Systematic Reviews with PROSPERO registration number: CRD42022314865.
Please change to:
This systematic review and meta-analysis was conducted in accordance with the ……

Validity of the findings

- Please mention about inclusion and exclusion criteria in this SEARCH STRATEGY section
- It is better if the authors mention in this section the search term that were used in this study.
- Inclusion and exclusion criteria should be write in Research strategy part.

Please write with no enumeration

- Please mention/explain what data were extracted from each article.
For example:
The following data were extracted from each article by two reviewers independently: study type, total number of participants……….

- Please combine all of these sections to one section: Characteristic of included studies
- In Discussion part, the authors should analyze the Results. Please do not state the results in Discussion part (Discussion does not repeat the results).

The authors should explain/mention the strength of their study.

Additional comments

-

Reviewer 3 ·

Basic reporting

This paper performed a systematic review and meta-analysis on the associations between depression and insomnia by 8 studies in Ethiopia, and its associated risk factors among healthcare professionals.


Major points:
Line 194-196: incorrect citation – e.g., didn’t rephrase, directly copy the exact text from another paper (word for word, but without quotation marks). -doi:10.1136/bmj.d4002
https://integrity.mit.edu/handbook/academic-writing/avoiding-plagiarism-paraphrasing

Line 256-273: main part of the discussion section was just simply replicating what authors had already mentioned in the result section.
Figure 2-7: incorrect scale of the plots.

Many spelling errors:
Typo/grammar: Line 32, Line 64, Line 94, Line 249, etc. (and also in authors’ “rationale” file)
Line 40-42: inconsistent font size

Minor points:
Line 58: spell out at first use
Figures: low resolution for publication.

Experimental design

The research design was insufficient.

Major points:
Line 32: did not specify up to which date.
Line 125: what specific sample size was considered as too small to be excluded?

Validity of the findings

Major points:
Line 217: three studies for insomnia were insufficient. (ref: doi: 10.1097/XEB.0000000000000065)
Line 236, 240, 242, 245, etc.: incorrect explanations on odds ratio.
Line 240, 246: I don’t think they are significant results.

Additional comments

I don’t think authors proofread the manuscript before they submitted it.

---

## Round 0.2 · Major Revisions

Please address the reviewers' comments.

Reviewer 1 ·

Basic reporting

The article is well written, but it needs improving in some parts after revising.
The keyword part is suddenly missing. The previous manuscript contained the keywords but in the revision part there is no keywords. The authors should obey the writing style according to journal policy (Line 52)
The authors must match the appropriate style while writing the reference(s). (Line 63-65)

Experimental design

Line Section Comment
52 Abstract The keyword part is missing. The previous manuscript before this revised manuscript still contained the keyword part. Please match with the writing style according to journal policy
63-65; 81-84 Introduction Does this style of writing the reference match with the requirement of the journal?
167-170 Results Please the authors to match with the style of the references writing according to this journal
246-248 Discussions Please the authors to match with the style of references writing according to the journal

Validity of the findings

-

Additional comments

-

·

Basic reporting

clear and unambiguous

Experimental design

1. as aims and scope this journal
2. methods described sufficient

Validity of the findings

all underlying data have been provided and controlled

Additional comments

can be considering to accepted and published

Reviewer 3 ·

Basic reporting

New comments were marked with **.

This paper performed a systematic review and meta-analysis on the associations between depression and insomnia by 8 studies in Ethiopia, and its associated risk factors among healthcare professionals.
Major points:
Line 194-196: incorrect citation – e.g., didn’t rephrase, directly copy the exact text from another paper (word for word, but without quotation marks). -doi:10.1136/bmj.d4002
https://integrity.mit.edu/handbook/academic-writing/avoiding-plagiarism-paraphrasing
*Thank you. For the comment, we modified sentence to avoid the similarity.

**Reviewer: Concerned some incorrect citations (it should not be checked by reviewers):
Line 118-119: “and expert opinions, not precisely measured the prevalence and the determinants of mental illness” – identical/similar wording to the original text “and expert opinions that have not precisely measured both the prevalence and the determinants of …”. (https://www.ncbi.nlm.nih.gov/pmc/articles/PMC8858466/)
Line 142 – 145: identical/similar wording to the original text “NOS scale rates observational studies based on 3 parameters: selection, comparability between the exposed and unexposed groups, and exposure/outcome assessment. …” (https://www.nature.com/articles/s41598-021-85359-3)


Line 256-273: main part of the discussion section was just simply replicating what authors had already mentioned in the result section.
* Thank you for the comments. We have modified the discussion part by incorporating the urgency of this study, analysis of results, comparison of the results with other countries and strength and limitation.
Figure 2-7: incorrect scale of the plots.
*We have re generated the plots for better clarity and resolution.

**Reviewer:
The scale of the plots was too small.

Many spelling errors:
Typo/grammar: Line 32, Line 64, Line 94, Line 249, etc. (and also in authors’ “rationale” file)
Line 40-42: inconsistent font size
Minor points:
Line 58: spell out at first use
*Thank you for the comments. We checked and modified the typo/grammar issues as much as possible.

* We have corrected the resolution for figures.

**Reviewer:
Minor point:
There are still many typos or inconsistent font:
Line 230, Line 233, Line 256, Line 271, Line 274, Table 4.

Experimental design

Experimental design
The research design was insufficient.
*The study is a systematic review of observational studies. We believe the design was appropriate for the study.
Major points:
Line 32: did not specify up to which date.
* Thank you for the comment. were searched to get literatures and articles published to June 2022 were included. We searched articles published from the occurence of the pandemic to June 2022 were included.
Line 125: what specific sample size was considered as too small to be excluded?
* We consider small sample size when n<30.

**Reviewer:
No further comments.

Validity of the findings

Major points:
Line 217: three studies for insomnia were insufficient.
*We tried to get more studies for a more-stronger evidence. But there exist lack of primary studies reporting insomnia. We have mentioned this in the limitation of the study.
(ref: doi: 10.1097/XEB.0000000000000065)

**Reviewer:
Major point:
Line 291-294: Please be more specific about the absence of sufficient studies (e.g., which mental illness was insufficient? How many studies did you have? How many studies should you have? etc.)


Line 236, 240, 242, 245, etc.: incorrect explanations on odds ratio.
Line 240, 246: I don’t think they are significant results.
*Thank you for the comments. In this study we pool the effect size (adjusted odds ratio) of the variables and we obtained sex (female) and maritial status (been married) are statisticaly significant. Whereas, working unit (isolation center) and with medical illness are not statisticaly significant variables on affecting the prevalence of depression. We also modified the explanations as well.

**Reviewer:
Major point:
Line 285-287; line 298-299: The results from Figure 6 and Figure 7 were the same. How did authors know if it was due to different regions or due to different instruments?

Additional comments

Overall, there was an improvement, but the evidence that authors provided was not strong. For example, there was no test to show whether there was a significant difference between the subgroups.

---

## Round 0.3 · accepted · Accept

Congratulations and I wish to see this paper online soon.